# DEEP BOOSTING OF DIVERSE EXPERTS

## ABSTRACT

In this paper, a deep boosting algorithm is developed to learn more discriminative ensemble classifier by seamlessly combining a set of base deep CNNs (base experts) with diverse capabilities, e.g., these base deep CNNs are sequentially trained to recognize a set of object classes in an easy-to-hard way according to their learning complexities. Our experimental results have demonstrated that our deep boosting algorithm can significantly improve the accuracy rates on large-scale visual recognition.

## 1 INTRODUCTION

The rapid growth of computational powers of GPUs has provided good opportunities for us to develop scalable learning algorithms to leverage massive digital images to train more discriminative classifiers for large-scale visual recognition applications, and deep learning (Simonyan & Zisserman, 2015; Szegedy et al., 2015; He et al., 2016) has demonstrated its outstanding performance because highly invariant and discriminant features and multi-way softmax classifier are learned jointly in an end-to-end fashion.

Before deep learning becomes so popular, boosting has achieved good success on visual recognition (Viola & Jones, 2004). By embedding multiple weak learners to construct an ensemble one, boosting (Schapire, 1999) can significantly improve the performance by sequentially training multiple weak learners with respect to a weighted error function which assigns larger weights to the samples misclassified by the previous weak learners. Thus it is very attractive to invest whether boosting can be integrated with deep learning to achieve higher accuracy rates on large-scale visual recognition.

By using neural networks to replace the traditional weak learners in the boosting frameworks, boosting of neural networks has received enough attentions (Zhou & Lai, 2009; Mosca & Magoulas, 2017a; Kuznetsov et al., 2014; Moghimi et al., 2016). All these existing deep boosting algorithms simply use the weighted error function (proposed by Adaboost (Schapire, 1999)) to replace the softmax error function (used in deep learning ) that treats all the errors equally. Because different object classes may have different learning complexities, it is more attractive to invest new deep boosting algorithm that can use different weights over various object classes rather than over different training samples.

Motivated by this observation, a deep boosting algorithm is developed to generate more discriminative ensemble classifier by combining a set of base deep CNNs with diverse capabilities, e.g., all these base deep CNNs (base experts) are sequentially trained to recognize different subsets of object classes in an easy-to-hard way according to their learning complexities. The rest of the paper is organized as: Section 2 briefly reviews the related work; Section 3 introduce our deep boosting algorithm; Section 4 reports our experimental results; and we conclude this paper at Section 5.

## 2 RELATED WORK

In this section, we briefly review the most relevant researches on deep learning and boosting.

Even deep learning has demonstrated its outstanding abilities on large-scale visual recognition (Krizhevsky et al., 2012; Simonyan & Zisserman, 2015; Szegedy et al., 2015; He et al., 2016; Huang et al., 2016), it still has room to improve: all the object classes are arbitrarily assumed to share similar learning complexities and a multi-way softmax is used to treat them equally. For recognizing large numbers of object classes, there may have significant differences on their learning complexi-

ties, e.g., some object classes may be harder to be recognized than others. Thus learning their deep CNNs jointly may not be able to achieve the global optimum effectively because the gradients of their joint objective function are not uniform for all the object classes and such joint learning process may distract on discerning some object classes that are hard to be discriminated. For recognizing large numbers of object classes with diverse learning complexities, it is very important to organize them in an easy-to-hard way according to their learning complexities and learn their deep CNNs sequentially. By assigning different weights to the training samples adaptively, boosting (Schapire, 1999; Freund & Schapire, 1997; Schapire et al., 1998) has provided an easy-to-hard approach to train a set of weak learners sequentially. Thus it is very attractive to invest whether we can leverage boosting to learn a set of base deep CNNs sequentially for recognizing large numbers of object classes in an easy-to-hard way.

Some deep boosting algorithms have been developed by seamlessly integrating boosting with deep neural networks to improve the performance in practice. Schwenk & Bengio (1997; 2000) proposed the first work to integrate Adaboost with neural networks for online character recognition application. Zhou & Lai (2009) extended the Adaboosting neural networks algorithm for credit scoring. Recently, Mosca & Magoulas (2017b) developed a deep incremental boosting method which increases the size of neural network at each round by adding new layers at the end of the network. Moreover, Mosca & Magoulas (2017a) integrated residual networks with incremental boosting and built an ensemble of residual networks via adding one more residual block to the previous residual network at each round of boosting. All these methods combine the merits of boosting and neural networks; they train each base network either using a different training set by resampling with a probability distribution derived from the error weight, or directly using the weighted cost function for the base network.

Alternatively, Saberian & Vasconcelos (2011) proposed a margin enforcing loss for multi-class boosting and presented two ways to minimize the resulting risk: the one is coordinate descent approach which updates one predictor component at a time, the other way is based on directional functional derivative and updates all components jointly. By applying the first way, i.e., coordinate descent, Cortes et al. (2014) designed ensemble learning algorithm for binary-class classification using deep decision trees as base classifiers and gave the data-dependent learning bound of convex ensembles, and Kuznetsov et al. (2014) furthermore extended it to multi-class version. By applying the second way, i.e., directional derivative descent, Moghimi et al. (2016) developed an algorithm for boosting deep convolutional neural networks (CNNs) based on least squares between weights and directional derivatives, which differs from the original method based on inner product of weights and directional derivative in (Saberian & Vasconcelos, 2011). All above algorithms focus on seeking the optimal ensemble predictor via changing the error weights of samples; they either update one component of the predictor per boosting iteration, or update all components simultaneously.

On the other hand, our deep boosting algorithm focuses on combining a set of base deep CNNs with diverse capabilities: (1) large numbers of object classes are automatically organized in an easy-to-hard way according to their learning complexities; (2) all these base deep CNNs (base experts) are sequentially learned to recognize different subsets of object classes; and (3) these base deep CNNs with diverse capabilities are seamlessly combined to generate more discriminative ensemble classifier.

## 3 DEEP BOOSTING OF DIVERSE EXPERTS

In this paper, a deep boosting algorithm is developed by seamlessly combining a set of base deep CNNs with various capabilities, e.g., all these base deep CNNs are sequentially trained to recognize different subsets of object classes in an easy-to-hard way according to their learning complexities. Our deep boosting algorithm uses the base deep CNNs as its weak learners, and many well-designed deep networks (such as AlexNet (Krizhevsky et al., 2012), VGG (Simonyan & Zisserman, 2015), ResNet (He et al., 2016), and huang2016densely), can be used as its base deep CNNs. It is worth noting that all these well-designed deep networks [] optimize their structures (i.e., numbers of layers and units in each layer), their node weights and their softmax jointly in an end-to-end manner for recognizing the same set of object classes. Thus our deep boosting algorithm is firstly implemented for recognizing 1,00 object classes, however, it is straightward to extend our current implementation

---

**Algorithm 1** Deep Boosting of Diverse Experts

---

**Require:** Training set from $C$ classes: $\{(\mathbf{x}_i, y_i) \mid y_i \in \{1, ..., C\}, i = 1, ..., N\}$; Initial significance distribution over categories: $[D_1(1), ..., D_1(C)]$; Number of base deep CNNs: $T$.

1: **for** $t = 1, ..., T$ **do**
2:     Normalization: $\widetilde{D}_t(l) = \frac{D_t(l)}{\sum_{j=1}^{C} D_t(j)}, (l = 1, ..., C)$
3:     Training the $t^{th}$ base deep CNNs $\mathbf{f}_t(\mathbf{x})$ via $Loss_t$ with respect to the importance distribution over C categories $[\widetilde{D}_t(1), ..., \widetilde{D}_t(C)]$;
4:     Calculating the error per category for $\mathbf{f}_t(\mathbf{x})$: $\varepsilon_t(l), (l = 1, ..., C)$;
5:     Computing the weighted error for $\mathbf{f}_t(\mathbf{x})$: $\varepsilon_t = \sum_{l=1}^{C} \widetilde{D}_t(l)\varepsilon_t(l)$;
6:     Setting $\beta_t = \frac{\lambda\varepsilon_t}{1 - \lambda\varepsilon_t}$;
7:     Updating $D_{t+1}(l)$ as $D_{t+1}(l) = D_t(l)\beta_t^{1-\lambda\varepsilon_t(l)}, (l = 1, ..., C)$, so that hard object classes misclassified by $\mathbf{f}_t(\mathbf{x})$ can receive larger weights (importances) when training the $(t+1)^{th}$ base deep CNNs at the next round;
8: **end for**
9: Ensembling: $\mathbf{g}(\mathbf{x}) = \frac{1}{\mathbb{Z}} \sum_{t=1}^{T} \log\left(\frac{1}{\beta_t}\right) \mathbf{f}_t(\mathbf{x})$

---

when huge deep networks (with larger capacities) are available in the future and being used as the base deep CNNs.

## 3.1 ALGORITHM

As illustrated in Algorithm 1, our deep boosting algorithm contains the following key components: (a) Training the $t^{th}$ base deep CNNs (base expert) $\mathbf{f}_t(\mathbf{x})$ by focusing on achieving higher accuracy rates for some particular object classes; (b) Estimating the weighted error function for the $t^{th}$ base deep CNNs $\mathbf{f}_t(\mathbf{x})$ according to the distribution of importances $D_t$ for $C$ object classes; (c) Updating the distribution of importances $D_{t+1}$ for $C$ object classes to train the $(t + 1)^{th}$ base deep CNNs by spending more efforts on distinguishing the hard object classes which are not classified very well by the previous base deep CNNs; (d) Such iterative training process stops when the maximum number of iterations is reached or a certain level of the accuracy rates is achieved.

For the $t^{th}$ base expert $\mathbf{f}_t(\mathbf{x})$, we firstly employ deep CNNs to map $\mathbf{x}$ into more separable feature space $\mathbf{h}_t(\mathbf{x}; \theta_t)$, followed by a fully connected discriminant layer and a C-way softmax layer. The output of the $t^{th}$ base expert is the predicted multi-class distribution, denoted as $\mathbf{f}_t(\mathbf{x}) = [p_t(1|\mathbf{x}), ..., p_t(C|\mathbf{x})]^\top$, whose each component $p_t(l|\mathbf{x})$ is the probability score of $\mathbf{x}$ assigned to the object class $l, (l = 1, ..., C)$:

$$p_t(l|\mathbf{x}) = \frac{exp\{\mathbf{w}_{lt}^\top \mathbf{h}_t(\mathbf{x}; \theta_t)\}}{\sum_{j=1}^{C} exp\{\mathbf{w}_{jt}^\top \mathbf{h}_t(\mathbf{x}; \theta_t)\}} \tag{1}$$

where $\theta_t$ and $\mathbf{w}_{lt}, (l = 1, ..., C)$ are the model parameters for the $t^{th}$ base expert $\mathbf{f}_t(\mathbf{x})$. Based on the above probability score, the category label of $\mathbf{x}$ can be predicted by the $t^{th}$ base expert as follows:

$$\hat{y}^t = \arg\max_l p_t(l|\mathbf{x}) \tag{2}$$

Suppose that training set consists of $N$ labeled samples from $C$ classes: $\{(\mathbf{x}_i, y_i) \mid y_i \in \{1, ..., C\}\}_{i=1}^{N}$. To train the $t^{th}$ base expert $\mathbf{f}_t(\mathbf{x})$, the model parameters can be learned by maximizing the objective function in the form of weighted margin as follows:

$$\mathcal{O}_t(\theta_t, \{\mathbf{w}_{lt}\}_{l=1}^{C}) = \sum_{l=1}^{C} \widetilde{D}_t(l)\xi_{lt} \tag{3}$$

where

$$\xi_{lt} = \frac{1}{N_l} \sum_{i=1}^{N} \mathbf{1}(y_i = l) \log p_t(l|\mathbf{x}_i) - \frac{1}{N - N_l} \sum_{i=1}^{N} \mathbf{1}(y_i \neq l) \log p_t(l|\mathbf{x}_i) \tag{4}$$

Herein the indicator function $\mathbf{1}(y_i = l)$ is equal to 1 if $y_i = l$; otherwise zero. $N_l$ denotes the number of samples belonging to the $l^{th}$ object class. $\widetilde{D}_t(l)$ is the normalized importance score for class $l$ in the $t^{th}$ base expert $\mathbf{f}_t(\mathbf{x})$. By using the distribution of importances $[\widetilde{D}_t(1), ..., \widetilde{D}_t(C)]$ to approximate the learning complexities for $C$ object classes, our deep boosting algorithm can push the current base deep CNNs to focus on distinguishing the object classes which are hard classified by the previous base deep CNNs, thus it can support an easy-to-hard solution for large-scale visual recognition.

$\xi_{lt}$ measures the margin between the average confidence on correctly classified examples and the average confidence on misclassified examples for the $l^{th}$ object class. If the second item in Eq.(4) is small enough and negligible, $\xi_{lt} \approx \frac{1}{N_l} \sum_{i=1}^{N} \mathbf{1}(y_i = l) \log p_t(l|\mathbf{x}_i)$, then maximizing the objective function in Eq.(3) is equivalent to maximizing the weighted likelihood.

For the $t^{th}$ base expert $\mathbf{f}_t(\mathbf{x})$, the classification error rate over the training samples in $l^{th}$ object class is as follows:

$$\epsilon_t(l) = \frac{1}{2} \sum_{i=1}^{N} \{ \mathbf{1}(y_i = l) \frac{1 - p_t(l|\mathbf{x}_i)}{N_l} + \mathbf{1}(y_i \neq l) \frac{p_t(l|\mathbf{x}_i)}{N - N_l} \} \tag{5}$$

This error rate is used to update category weight and the loss function of the next weak learner, and above definition encourages predictors with large margin to improve the discrimination between correct class and incorrect classe competing with it. Error rate calculated by Eq.(6) is in soft decision with probability; alternatively, we can also simply compute the error rate in hard decision as $\epsilon_t(l) = \frac{1}{N_l} \sum_{i=1}^{N} \mathbf{1}(y_i = l \wedge \hat{y}_i^t \neq l)$. The criterion for hard object classes is $\epsilon_t(l) > \frac{1}{2\lambda}$ where the hyper-parameter $\lambda$ controls the threshold, and we constrain $\lambda > \frac{1}{2}$ (i.e., $\frac{1}{2\lambda} < 1$ ) such that the threshold makes sense. The larger the hyper-parameter $\lambda$ is, the more strict the precision requirement becomes. We then compute the weighted error rate $\varepsilon_t$ over all classes for $\mathbf{f}_t(\mathbf{x})$ such that hard object classes are focused on by the next base expert.

$$\varepsilon_t = \sum_{l=1}^{C} \widetilde{D}_t(l) \epsilon_t(l) \tag{6}$$

The distribution of importances is initialized equally for all $C$ object classes: $D_1(l) = \frac{1}{C}, (l = 1, ..., C)$, and it is updated along the iterative learning process by emphasizing the object classes which are heavily misclassified by the previous base deep CNNs:

$$D_{t+1}(l) = D_t(l) \beta_t^{1 - \lambda \epsilon_t(l)} \tag{7}$$

where $\beta_t$ should be an increasing function of $\varepsilon_t$, and its range should be $0 < \beta_t < 1$. It should be pointed out that $\lambda \epsilon_t(l)$ denotes the product of $\lambda$ and $\epsilon_t(l)$. Such update of distribution encourages the next base network focusing on the categories that are hard to classify. As shown in **Section 4**, to guarantee the upper boundary of ratio (the number of heavily misclassified categories over the number of all classes) to be minimized, we set

$$\beta_t = \frac{\lambda \varepsilon_t}{1 - \lambda \varepsilon_t} \tag{8}$$

Normalization of the updated importances distribution can be easily carried out:

$$\widetilde{D}_{t+1}(l) = \frac{D_{t+1}(l)}{\sum_{j=1}^{C} D_{t+1}(j)}, \qquad (l = 1, ..., C) \tag{9}$$

The distribution of importances is used to: (a) separate the hard object classes (heavily misclassified by the previous base deep CNNs) from the easy object classes (which have been classified correctly by the previous base deep CNNs); (b) estimate the weighted error function for the $(t + 1)^{th}$ base deep CNNs $\mathbf{f}_{t+1}(\mathbf{x})$, so that it can spend more efforts on distinguishing the hard object classes misclassified by the previous base deep CNNs.

After $T$ iterations, we can obtain $T$ base deep CNNs (base experts) $\{\mathbf{f}_1, \cdots, \mathbf{f}_t, \cdots, \mathbf{f}_T\}$, which are sequentially trained to recognize different subsets of $C$ object classes in an easy-to-hard way according to their learning complexities. All these $T$ base deep CNNs are seamlessly combined to generate more discriminative ensemble classifier $\mathbf{g}(\mathbf{x})$ for recognizing $C$ object classes:

$$\mathbf{g}(\mathbf{x}) = \frac{1}{\mathbb{Z}} \sum_{t=1}^{T} \log\left(\frac{1}{\beta_t}\right) \mathbf{f}_t(\mathbf{x}) \tag{10}$$

where $\mathbb{Z} = \sum_{t=1}^{T} \log\left(\frac{1}{\beta_t}\right)$ is a normalization factor. By diversifying a set of base deep CNNs on their capabilities (i.e., they are trained to recognize different subsets of $C$ object classes in an easy-to-hard way), our deep boosting algorithm can obtain more discriminative ensemble classifier $\mathbf{g}(\mathbf{x})$ to significantly improve the accuracy rates on large-scale visual recognition.

To apply such ensembled classifier for recognition, for a given test sample $\mathbf{x}_{test}$, it firstly goes through all these base deep CNNs to obtain $T$ deep representations $\{\mathbf{h}_1, \cdots, \mathbf{h}_T\}$ and then its final probability score $p(l|\mathbf{x}_{test})$ to be assigned into the $l$th object class is calculated as follows:

$$p(l|\mathbf{x}_{test}) = \frac{1}{\mathbb{Z}} \sum_{t=1}^{T} \log\left(\frac{1}{\beta_t}\right) \frac{exp\{\mathbf{w}_{lt}^{\top} \mathbf{h}_t(\mathbf{x}_{test}; \theta_t)\}}{\sum_{j=1}^{C} exp\{\mathbf{w}_{jt}^{\top} \mathbf{h}_t(\mathbf{x}_{test}; \theta_t)\}} \tag{11}$$

## 3.2 SELECTION OF $\beta_t$

In our deep boosting algorithm, $\beta_t$ is selected to be an increasing function of error rate $\varepsilon_t$, with its range $[0, 1]$. $\beta_t$ is employed in two folds: (i) As seen in Eq.(7), $\beta_t$ helps to update the importance of different categories such that hard object classes are emphasized; (ii) As seen in Eq.(10) and Eq.(11), reciprocals of $\beta_t$ are the combination coefficients for the final ensemble classifier such that those base experts with low error rate have large weight.

The criterion of hard object classes for the $t$th expert is $\epsilon_t(l) > \frac{1}{2\lambda}$. Denote $\epsilon_{min}(l) \triangleq \min\{\epsilon_1(l), ..., \epsilon_T(l)\}$. If $\epsilon_{min}(l) > \frac{1}{2\lambda}$, then $\epsilon_t(l) > \frac{1}{2\lambda}$ for each $t$, $(t = 1, ..., T)$; it implies that the $l$th object class is hard for all $T$ experts.

Let $\sharp\{l : \epsilon_{min}(l) > \frac{1}{2\lambda}\}$ denote the the number of hard object classes for all $T$ experts. Inspired by Freund & Schapire (1997), we now show that the selection of $\beta_t$ as in Eq.(8) guarantees the upper boundary of ratio (the number of heavily misclassified categories over the number of all classes) to be minimized.

It can be shown that for $0 < x < 1$ and $0 < \alpha < 1$, we have $x^\alpha \le 1 - (1-x)\alpha$. According to Eq.(7):

$$D_{t+1}(l) = D_t(l)\beta_t^{1-\lambda\epsilon_t(l)},$$

we get

$$\sum_{l=1}^{C} D_{t+1}(l) = \sum_{l=1}^{C} D_t(l)\beta_t^{1-\lambda\epsilon_t(l)} \le \sum_{l=1}^{C} D_t(l)(1 - (1-\beta_t)(1 - \lambda\epsilon_t(l)))$$
$$= (\sum_{l=1}^{C} D_t(l))(1 - (1-\beta_t)) + \lambda(1-\beta_t)\sum_{l=1}^{C} D_t(l)\epsilon_t(l) \tag{12}$$

According to Eq.(6) and Eq.(9), we get:

$$\sum_{l=1}^{C} D_t(l)\epsilon_t(l) = (\sum_{l=1}^{C} D_t(l))\varepsilon_t$$

Therefore

$$\sum_{l=1}^{C} D_{t+1}(l) \le \sum_{l=1}^{C} D_t(l)(1 - (1-\beta_t)) + \lambda(1-\beta_t)(\sum_{l=1}^{C} D_t(l))\varepsilon_t$$
$$= (\sum_{l=1}^{C} D_t(l))[1 - (1-\beta_t)(1 - \lambda\varepsilon_t)] \tag{13}$$

Since $\sum_{l=1}^{C} D_1(l) = 1$, we have

$$\sum_{l=1}^{C} D_2(l) \leq (\sum_{l=1}^{C} D_1(l))[1 - (1 - \beta_1)(1 - \lambda\varepsilon_1)] = 1 - (1 - \beta_1)(1 - \lambda\varepsilon_1)$$

Thus

$$\sum_{l=1}^{C} D_{T+1}(l) \leq \Pi_{t=1}^{T}[1 - (1 - \beta_t)(1 - \lambda\varepsilon_t)] \tag{14}$$

By substituting Eq.(7) into Eq.(14), we get

$$\Pi_{t=1}^{T}[1 - (1 - \beta_t)(1 - \lambda\varepsilon_t)] \geq \sum_{l=1}^{C} D_{T+1}(l) = \sum_{l=1}^{C}(D_1(l)\Pi_{t=1}^{T}\beta_t^{1-\lambda\epsilon_t(l)})$$

$$= \frac{1}{C}\sum_{l=1}^{C}(\Pi_{t=1}^{T}\beta_t^{1-\lambda\epsilon_t(l)}) \geq \frac{1}{C}\sum_{l:\epsilon_{min}(l)>\frac{1}{2\lambda}}(\Pi_{t=1}^{T}\beta_t^{1-\lambda\epsilon_t(l)}) \tag{15}$$

Due to $\epsilon_{min}(l) > \frac{1}{2\lambda}$, it holds that $\epsilon_t(l) > \frac{1}{2\lambda}$ and $1 - \lambda\epsilon_t(l) < \frac{1}{2}$ for all $l$. Recall the constraint that $0 < \beta_t < 1$, thus

$$\frac{1}{C}\sum_{l:\epsilon_{min}(l)>\frac{1}{2\lambda}}(\Pi_{t=1}^{T}\beta_t^{1-\lambda\epsilon_t(l)}) \geq \frac{1}{C}\sum_{l:\epsilon_{min}(l)>\frac{1}{2\lambda}}(\Pi_{t=1}^{T}\beta_t^{\frac{1}{2}})$$

$$= \frac{\sharp\{l : \epsilon_{min}(l) > \frac{1}{2\lambda}\}}{C}\Pi_{t=1}^{T}\beta_t^{\frac{1}{2}} \tag{16}$$

Combining Eq.(15) with Eq.(16), we get

$$\frac{\sharp\{l : \epsilon_{min}(l) > \frac{1}{2\lambda}\}}{C} \leq \frac{\Pi_{t=1}^{T}[1 - (1 - \beta_t)(1 - \lambda\varepsilon_t)]}{\Pi_{t=1}^{T}\beta_t^{\frac{1}{2}}}$$

$$= \Pi_{t=1}^{T}\frac{1 - (1 - \beta_t)(1 - \lambda\varepsilon_t)}{\beta_t^{\frac{1}{2}}} \tag{17}$$

To minimize the rightside, we set its partial derivative with respect to $\beta_t$ to zero:

$$\frac{\partial}{\partial\beta_t}(\Pi_{t=1}^{T}\frac{1 - (1 - \beta_t)(1 - \lambda\varepsilon_t)}{\beta_t^{\frac{1}{2}}}) = 0$$

Since $\beta_t$ only exists in the $t$th factor, above equation is equivalent to

$$\frac{\partial}{\partial\beta_t}(\frac{1 - (1 - \beta_t)(1 - \lambda\varepsilon_t)}{\beta_t^{\frac{1}{2}}}) = 0$$

Solving it, we find that $\beta_t$ can be optimally selected as:

$$\beta_t = \frac{\lambda\varepsilon_t}{1 - \lambda\varepsilon_t}$$

### 3.3 SELECTION OF $\lambda$

We substitute $\beta_t = \frac{\lambda\varepsilon_t}{1-\lambda\varepsilon_t}$ into Eq.(17), and get the upper boundary of ratio (the number of hard object categories over the number of all classes):

$$\frac{\sharp\{l : \epsilon_{min}(l) > \frac{1}{2\lambda}\}}{C} \leq 2^T\Pi_{t=1}^{T}\sqrt{\lambda\varepsilon_t(1 - \lambda\varepsilon_t)} \tag{18}$$

Now we discuss the range for the hyper-parameter $\lambda$. Recall that the criterion of hard object classes for the $t$th expert is $\epsilon_t(l) > \frac{1}{2\lambda}$ and $\lambda$ controls the threshold. When we are to raise the precision

requirements, $\lambda$ can be set large. We constrain $\lambda > \frac{1}{2}$ such that $\frac{1}{2\lambda} < 1$ and the threshold of error rate makes sense. On the other hand, the range of $\beta_t = \frac{\lambda\varepsilon_t}{1-\lambda\varepsilon_t}$ is $0 < \beta_t < 1$, so it is required that $\lambda\varepsilon_t < \frac{1}{2}$, i.e., $\lambda < \frac{1}{2\varepsilon_t}$. To conclude, $\lambda$ should be selected between the interval $[\frac{1}{2}, \frac{1}{2\varepsilon_t}]$.

From the relation between $\lambda\varepsilon_t$ and $\lambda\varepsilon_t(1 - \lambda\varepsilon_t)$, as illustrated in Fig.1, we can see the effect of $\lambda$ on the upper boundary of ratio (the number of hard object categories over the number of all classes) in Eq.(18).

- In the yellow shaded region, $\lambda \in [\frac{1}{2}, \frac{1}{2\varepsilon_t}]$, i.e., $\frac{\varepsilon_t}{2} < \lambda\varepsilon_t < \frac{1}{2}$, the condition $0 < \beta_t < 1$ is satisfied, and the upper boundary of hard category percentage in Eq.(18) increases with $\lambda$ increasing, the reason for which is that when $\lambda$ increases, the precision requirement increases, thus the number of hard categories increases too.

- On the right side of the yellow shaded region, $\lambda > \frac{1}{2\varepsilon_t}$, i.e., $\lambda\varepsilon_t > \frac{1}{2}$. In this case, the condition $0 < \beta_t = \frac{\lambda\varepsilon_t}{1-\lambda\varepsilon_t} < 1$ is not satisfied, thus the update of importance distribution in Eq.(7) can not effectively emphasize the object classes which are heavily misclassified by the previous experts. In hard classification task, large error rates $\varepsilon_t$ tend to result in $\lambda\varepsilon_t$ larger than or approaching $\frac{1}{2}$, and $\beta_t$ larger than or approaching 1. The value of $\lambda$ should be set smaller to alleviate large $\varepsilon_t$ such that $\lambda\varepsilon_t < \frac{1}{2}$ and $0 < \beta_t < 1$.

- On the left side of the yellow shaded region, $\lambda < \frac{1}{2}$, i.e., $\frac{1}{2\lambda} > 1$, then it can not be used as the the criterion of hard object classes that $\epsilon_t(l) > \frac{1}{2\lambda}$.

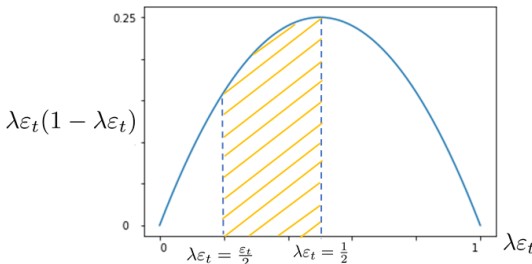

Figure 1: The relation between $\lambda\varepsilon_t$ and $\lambda\varepsilon_t(1-\lambda\varepsilon_t)$. The domain with respect to $\lambda$ is $\frac{1}{2} < \lambda < \frac{1}{2\varepsilon_t}$, so $\frac{\varepsilon_t}{2} < \lambda\varepsilon_t < \frac{1}{2}$ (yellow shaded region).

## 3.4 BACK PROPAGATION

The procedure of learning the $t^{th}$ base expert repeatedly adjusts the parameters of the corresponding deep network so as to maximize the objective function $\mathcal{O}_t$ in Eq.(3). To maximize $\mathcal{O}_t$, it is necessary to calculate its gradients with respect to all parameters, including the weights $\{\mathbf{w}_{lt}\}_{l=1}^{C}$ and the set of model parameters $\theta_t$.

For clearance, we denote

$$z_{lt}(\mathbf{x}) = \mathbf{w}_{lt}^{\top}\mathbf{h}_t(\mathbf{x}; \theta_t) \triangleq z_{lt}(\mathbf{h}_t, \mathbf{w}_{lt})$$

Thus, the probability score of $\mathbf{x}$ assigned to the object class $l$, $(l = 1, ..., C)$, in Eq.(1) can be written as

$$p_t(l|\mathbf{x}) = \frac{e^{z_{lt}(\mathbf{x})}}{\sum_{j=1}^{C} e^{z_{jt}(\mathbf{x})}} \triangleq p_t^l(z_{1t}, ..., z_{Ct})$$

Then, the objective function in Eq.(3) can be denoted as

$$\mathcal{O}_t(\theta_t, \{\mathbf{w}_{lt}\}_{l=1}^{C}) \triangleq \mathcal{O}_t(p_t^1, ..., p_t^C)$$

From above presentations, it can be more clearly seen that the objective is a composite function. $\mathcal{O}_t$ is firstly a function of multiple variables $p_t^1, ..., p_t^C$, whose each component $p_t^l$ is a function of

multiple variables $z_{1t}, ..., z_{Ct}$, and each $z_{lt}$ is a function of multiple variables $\mathbf{h}_t, \mathbf{w}_{lt}$, furthermore, each $\mathbf{h}_t$ is a function of multiple variables $\theta_t$. By chain rule, the gradients for the objective function $\mathcal{O}_t$ with respect to $\{\mathbf{w}_{lt}\}_{l=1}^C$ and $\theta_t$ are computed from the top layer to the bottom one in backward pass.

$$\frac{\partial \mathcal{O}_t}{\partial \mathbf{w}_{lt}} = \sum_{j=1}^C \frac{\partial \mathcal{O}_t}{\partial p_t^j} \frac{\partial p_t^j}{\partial z_{lt}} \frac{\partial z_{lt}}{\partial \mathbf{w}_{lt}}, \qquad \frac{\partial \mathcal{O}_t}{\partial \theta_t} = \sum_{j,l=1}^C \frac{\partial \mathcal{O}_t}{\partial p_t^j} \frac{\partial p_t^j}{\partial z_{lt}} \frac{\partial z_{lt}}{\partial \mathbf{h}_t} \frac{\partial \mathbf{h}_t}{\partial \theta_t}$$

where

$$\frac{\partial \mathcal{O}_t}{\partial p_t^j} = \widetilde{D}_t(j)[\frac{1}{N_j} \sum_{i=1}^N \mathbf{1}(y_i = j) \frac{1}{p_t(j|\mathbf{x}_i)} - \frac{1}{N - N_j} \sum_{i=1}^N \mathbf{1}(y_i \neq j) \log p_t(j|\mathbf{x}_i)]$$

$$\frac{\partial p_t^j}{\partial z_{lt}} = \left\{ \begin{array}{ll} p_t^l(1 - p_t^l) & \text{if } j = l \\ -p_t^l p_t^j & \text{if } j \neq l \end{array} \right.$$

and

$$\frac{\partial z_{lt}}{\partial \mathbf{w}_{lt}} = \mathbf{h}_t, \qquad \frac{\partial z_{lt}}{\partial \mathbf{h}_t} = \mathbf{w}_{lt}, \qquad \frac{\partial \mathbf{h}_t}{\partial \theta_t} = J$$

Herein, $J$ is Jacobi matrix. Such gradients are back-propagated [] through the $t^{th}$ base deep CNNs to fine-tune the weights $\{\mathbf{w}_{lt}\}_{l=1}^C$ and the set of model parameters $\theta_t$ simultaneously.

## 3.5 Generalization Error Bound

Denote $\mathcal{X}$ as the instance space, denote $\Omega$ as the distribution over $\mathcal{X}$, and denote $\mathcal{S}$ as a training set of $N$ examples chosen i.i.d according to $\Omega$. We are to investigate the gap between the generalization error on $\Omega$ and the empirical error on $\mathcal{S}$.

Suppose that $\mathcal{F}$ is the set from which the base deep experts are chosen, and let $\mathcal{G} = \left\{\mathbf{x} \mapsto \sum_{\mathbf{f} \in \mathcal{F}} a_f \mathbf{f}(\mathbf{x})|a_f \geqslant 0, \sum_{\mathbf{f} \in \mathcal{F}} a_f = 1\right\}$. The combination coefficients $\mathcal{A} = [a_1, ..., a_f, ...]$ can be viewed as a distribution over $\mathcal{F}$. Define $\hat{\mathcal{G}} = \left\{\mathbf{x} \mapsto \frac{1}{\Gamma} \sum_{t=1}^\Gamma \mathbf{f}_t(x)|\mathbf{f}_t \in \mathcal{F}\right\}$, and each $\mathbf{f}_t \in \mathcal{F}$ may appear multiple times in the sum. For any $\mathbf{g} \in \mathcal{G}$, there exists a distribution $\mathcal{A} = [a_1, ..., a_f, ...]$, so we can select the base deep experts from $\mathcal{F}$ for $\Gamma$ times independently according to $\mathcal{A}$ and obtain $\hat{\mathbf{g}} \in \hat{\mathcal{G}}$, denote $\hat{g} \sim \mathcal{A}$.

Note that $\mathbf{g}$ is a $C$-dim vetor, and each component of $\mathbf{g}$ is the category confidence, i.e., $g_y(\mathbf{x}) = p(y|\mathbf{x}), (y = 1, ..., C)$. Based on Eq.(11), the category label of test sample can be predicted by $\arg\max_y g_y(\mathbf{x}) = p(y|\mathbf{x})$. The ensembled classifier $\mathbf{g}$ predicts wrong if $g_y(\mathbf{x}) \leq \max_{\bar{y} \neq y} g_{\bar{y}}(\mathbf{x})$. The generalization error rate for the final ensembled classifier can be measured by the probability $\mathbf{P}_\Omega[g_y(\mathbf{x}) \leq \max_{\bar{y} \neq y} g_{\bar{y}}(\mathbf{x})]$.

According to probability theory, for any events $B_1$ and $B_2$, $\mathbf{P}(B_1) \leq \mathbf{P}(B_2) + \mathbf{P}(\bar{B}_2|B_1)$, therefore

$$\begin{aligned} \mathbf{P}_\Omega[g_y(\mathbf{x}) \leq \max_{\bar{y} \neq y} g_{\bar{y}}(\mathbf{x})] \leq & \mathbf{P}_{\Omega, \hat{g} \sim \mathcal{A}}[\hat{g}_y(\mathbf{x}) \leq \max_{\bar{y} \neq y} \hat{g}_{\bar{y}}(\mathbf{x}) + \frac{\xi}{2}] + \\ & \mathbf{P}_{\Omega, \hat{g} \sim \mathcal{A}}[\hat{g}_y(\mathbf{x}) > \max_{\bar{y} \neq y} \hat{g}_{\bar{y}}(\mathbf{x}) + \frac{\xi}{2}|g_y(\mathbf{x}) \leq \max_{\bar{y} \neq y} g_{\bar{y}}(\mathbf{x})] \end{aligned} \tag{19}$$

where $\xi > 0$ measures the margin between the confidences from ground-truth and incorrect categories. Using Chernoff bound (Schapire et al., 1998), the the second term in the right side of Eq.(19) is bounded as:

$$\mathbf{P}_{\Omega, \hat{g} \sim \mathcal{A}}[\hat{g}_y(\mathbf{x}) > \max_{\bar{y} \neq y} \hat{g}_{\bar{y}}(\mathbf{x}) + \frac{\xi}{2}|g_y(\mathbf{x}) \leq \max_{\bar{y} \neq y} g_{\bar{y}}(\mathbf{x})] \leq e^{-\Gamma \xi^2 / 8} \tag{20}$$

Assume that the base-classifier space $\mathcal{F}$ is with VC-dimension $d$, which can be approximately estimated by the number of neurons $\nu$ and the number of weigths $\omega$ in the base deep network, i.e., $d = O(\nu\omega)$. Recall that $\mathcal{S}$ is a sample set of $N$ examples from $C$ categories. Then the effective number of hypotheses for $\mathcal{F}$ over $\mathcal{S}$ is at most $\sum_{i=1}^{d} \binom{CN}{i} = (\frac{eNC}{d})^d$. Thus, the effective number of hypotheses over $\mathcal{S}$ for $\hat{\mathcal{G}} = \left\{ \mathbf{x} \mapsto \frac{1}{\Gamma} \sum_{t=1}^{\Gamma} \mathbf{f}_t(x) | \mathbf{f}_t \in \mathcal{F} \right\}$ is at most $(\frac{eNC}{d})^{\Gamma d}$.

Applying Devroye Lemma as in (Schapire et al., 1998), it holds with probability at least $1 - \delta_\Gamma$ that

$$\mathbf{P}_{\Omega, \hat{g} \sim \mathcal{A}}[\hat{g}_y(\mathbf{x}) \leq \max_{\bar{y} \neq y} \hat{g}_{\bar{y}}(\mathbf{x}) + \frac{\xi}{2}] \leq \mathbf{P}_{\mathcal{S}, \hat{g} \sim \mathcal{A}}[\hat{g}_y(\mathbf{x}) \leq \max_{\bar{y} \neq y} \hat{g}_{\bar{y}}(\mathbf{x}) + \frac{\xi}{2}] + \Delta_\Gamma \qquad (21)$$

where $\Delta_\Gamma = \sqrt{\frac{1}{2N}[\Gamma d \log \frac{eN^2 C}{d} + \log \frac{4e^8(\Gamma+1)}{\delta_\Gamma}]}$.

Likewise, in probability theory for any events $B_1$ and $B_2$, $\mathbf{P}(B_1) \leq \mathbf{P}(B_2) + \mathbf{P}(B_1 | \bar{B}_2)$, thus

$$\mathbf{P}_{\mathcal{S}, \hat{g} \sim \mathcal{A}}[\hat{g}_y(\mathbf{x}) \leq \max_{\bar{y} \neq y} \hat{g}_{\bar{y}}(\mathbf{x}) + \frac{\xi}{2}] \leq \mathbf{P}_{\mathcal{S}}[g_y(\mathbf{x}) \leq \max_{\bar{y} \neq y} g_{\bar{y}}(\mathbf{x}) + \xi] +$$
$$\mathbf{P}_{\mathcal{S}, \hat{g} \sim \mathcal{A}}[\hat{g}_y(\mathbf{x}) \leq \max_{\bar{y} \neq y} \hat{g}_{\bar{y}}(\mathbf{x}) + \frac{\xi}{2} | g_y(\mathbf{x}) > \max_{\bar{y} \neq y} g_{\bar{y}}(\mathbf{x}) + \xi]$$
$$(22)$$

Because

$$\mathbf{P}_{\mathcal{S}, \hat{g} \sim \mathcal{A}}[\hat{g}_y(\mathbf{x}) \leq \max_{\bar{y} \neq y} \hat{g}_{\bar{y}}(\mathbf{x}) + \frac{\xi}{2} | g_y(\mathbf{x}) > \max_{\bar{y} \neq y} g_{\bar{y}}(\mathbf{x}) + \xi]$$
$$\leq \sum_{\bar{y} \neq y} \mathbf{P}_{\mathcal{S}, \hat{g} \sim \mathcal{A}}[\hat{g}_y(\mathbf{x}) \leq \hat{g}_{\bar{y}}(\mathbf{x}) + \frac{\xi}{2} | g_y(\mathbf{x}) > g_{\bar{y}}(\mathbf{x}) + \xi] \leq (C-1)e^{-\Gamma \xi^2 / 8} \qquad (23)$$

So, combining Eq.$(19-23)$ together, it can be derived that

$$\mathbf{P}_\Omega[g_y(\mathbf{x}) \leq \max_{\bar{y} \neq y} g_{\bar{y}}(\mathbf{x})] \leq \mathbf{P}_{\mathcal{S}}[g_y(\mathbf{x}) \leq \max_{\bar{y} \neq y} g_{\bar{y}}(\mathbf{x}) + \xi] + Ce^{-\Gamma \xi^2 / 8} + \Delta_\Gamma \qquad (24)$$

As can be seen from above, large margin $\xi$ over the training set corresponds to narrow gap between the generalization error on $\Omega$ and the empirical error on $\mathcal{S}$, which leads to the better upper bound of generalization error.

## 4    EXPERIMENTAL RESULTS

In this section we evaluate the proposed algorithms on three real world datasets MNIST (LeCun, 1998), CIFAR-100 (Krizhevsky & Hinton, 2009), and ImageNet (Russakovsky et al., 2015). For MNIST and CIFAR-100, we train all networks from scrach in each AdaBoost iteration stage. On ImageNet, we use the pretrained model as the result of iteration #1 and then train weighted models sequentially. The pretrained model is available in TorchVision[1]. In each iteration, we adopt the weight initialization menthod proposed by He et al. (2015). All the networks are trained using stochastic gradient descent (SGD) with the weight decay 0f $10^{-4}$ and the momentum of 0.9 in the experiments.

### EXPERIMENTAL RESULTS ON MNIST

MNIST dataset consists of 60,000 training and 10,000 test handwritten digit samples. Schwenk & Bengio (2000) showed the accuracy improvement of MLP via AdaBoost on MNIST dataset by updating sample weights according to classification errors. For fair comparison, we firstly use the similar network architecture (MLP) as the base experts in experiments. We train two sets of networks with the only difference that one updates weights w.r.t the class errors on training datasets while the other one updates weights w.r.t the sample errors on training datasets. The former is the proposed

---

[1]https://github.com/pytorch/vision

method in this paper, and the latter is the traditional AdaBoost method. In the two sets of weak learners, we share the same weak learner in iteration #1 and train other two weak learners seperately. For data pre-processing, we normalize data via subtracting means and dividing standard deviations. In the experiment on MNIST, we simply train the network with learning rate 0.01 through out the whole 120 epoches.

| Method | Iteration #1 | Iteration #2 | Iteration #3 |
|---|---|---|---|
| update weights w.r.t sample errors | 4.73 | 2.53 | 2.23 |
| update weights w.r.t class errors | 4.73 | 2.22 | 1.87 |

Table 1: Comparison of test error rate(%) with boosted model on MNIST datasets.

With our proposed method, the top 1 error on test datasets decreases from 4.73% to 1.87 % after three iterations (table 1). After the interation #1, the top 1 error of our method drops more quickly than the method which update weights w.r.t sample errors. Our method, which updates weights w.r.t the class errors, leverages the idea that different class should have different learning comlexity and should not be treated equally. Through the iterations, our method trains a set of classifiers in an easy-to-hard way.

Class APs vary from each weak learner in each iteration to others, increasing for marjor weighted classes while decreasing for minor weighted classes(fig. 2-left). Therefore, in each iteration, the weighted learner classifier behaves like a expert different from the classfier in the previous iteration. Though some APs for certain classes may decrease in some degree with each weak learner, the boosting models improve the accuracy for hard classes while preservering the accuracy for easy classes (fig. 2-right). Our method cordinates the set of weak learners trained sequeentially with diversified capabilities to improve the classfication capability of boosting model.

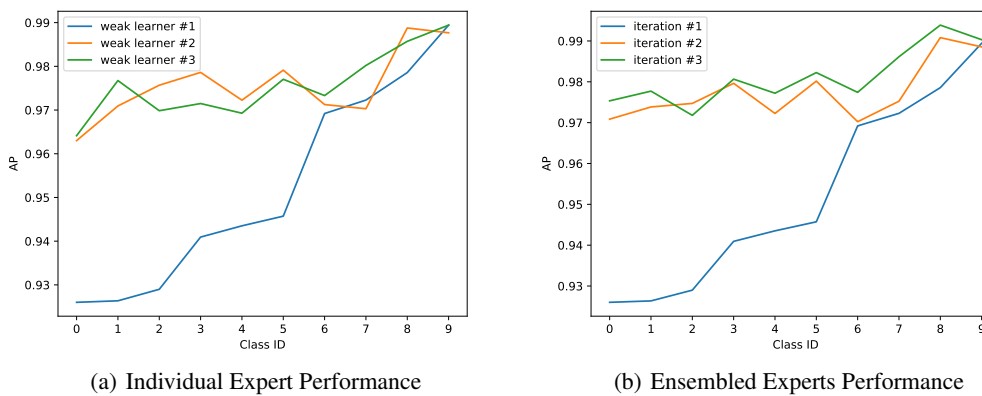

(a) Individual Expert Performance        (b) Ensembled Experts Performance

Figure 2: The comparison of class AP(average precision) on MNIST with MLP. The x-axis (class ID) is sorted according to the test APs (average pricision) per class in the weak learner #1. AP(average pricision) on the left plot is calculated on validation darasets with single model. AP (average pricision) on the right plot is calcucalted on test datasets with boosted model as in eq. (11) with $t=\{1,2,3\}$ respectively.

EXPERIMENTAL RESULTS ON CIFAR-100

We also carry out experiments on CIFAR-100 dataset. CIFAR-100 dataset consists of 60,000 images from 100 classes. There are 500 training images and 100 testing images per class. We adopt padding, mirroring, shifting for data augumentation and normalization as in (He et al., 2016; Huang et al., 2016). In training stage, we hold out 5,000 training images for validation and leave 45,000 for training. Because the error per class on training datasets approaches zero and training errors could be all zeros with even simple networks (Zhang et al., 2016), we update the category distribution w.r.t

the class errors on validation datasets. We do not use any sample of validation datasets to update parameters of the networks itself. When training networks on CIFAR-100, the initial learning rate is set to 0.1 and divided by 0.1 at epoch [150, 225]. Similar to (He et al., 2015; Huang et al., 2016), we train the network for 300 epoches. We show the results with various models including ResNet56($\lambda = 0.7$) and DenseNet-BC(k=12)(Huang et al., 2016) on test set. The performances of emsembled classifier with different number of base networks are shown in the middle two rows of (table 2).

As illustrated in section 3.3, $\lambda$ controls the weight differences among classes. In comparison, we use $\lambda$={0.7, 0.5, 0.1}. As shown in fig. 3-left, with smaller lambda, the weitht differences become bigger. We use ResNet model in (He et al., 2016) with 56 layers on CIFAR-100 datasets.

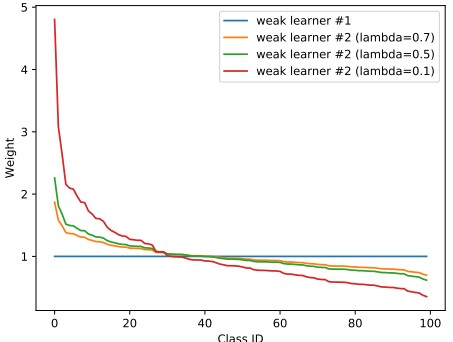 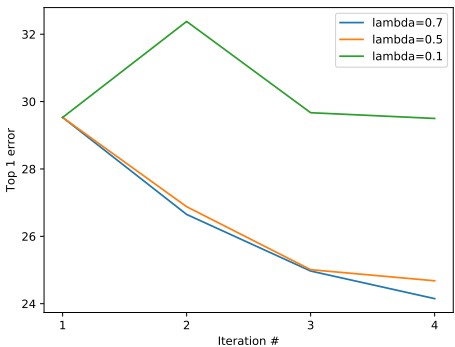

Figure 3: Comparison of lambda selection on CIFAR-100 datsets with ResNet56 model. Class IDs are sorted w.r.t the class APs of iteration #1. Top 1 errors (left plot) are generated on test datasets.

Overall, the models with lambda=0.7 performs the best, resulting in 24.15% test error after four iterations. Comparing with lambda=0.5 and lambda=0.7, we find that both model performs well in the initial several iterations, but the model with lambda=0.7 would converge to a better optimal(fig. 3-right). However, with lambda=0.1 which weights classes more discriminately, top 1 error fluctuates along the iterations (fig. 3-right). We conclude that lambda should be merely used to insure that the value of $\beta$ is below 0.5 and may harm the performance in the ensemble models if set to a low value.

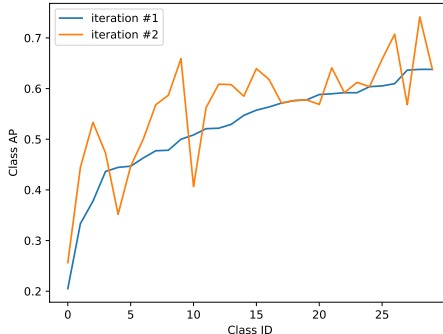 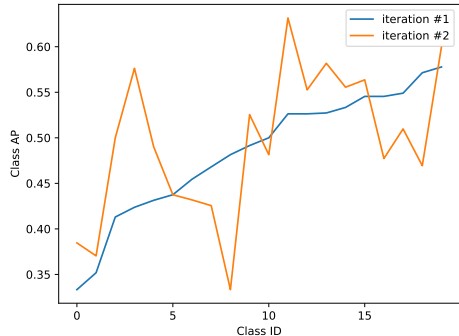

Figure 4: The change of validation APs from the classes with low APs in the iteration # 1 for ResNet56 (left) and DenseNet-BC(k=12) (right) on CIFAR-100. APs are all calculated from validation sets which would be used for the update of weights in the proceeding iteration. Class IDs are sorted by the APs of weak learner #1 and selected for those with small class IDs.

In fig. 4, we show the comparison of weak leaner #1 and weak learner #2 without boosting. Though with minor exeptions, most classes with low APs improve their class APs in the proceeding weak

learner. Our method is based on the motivation that different classes should have different learning comlexity. Thus, those classes with higher learning complexity should be paid more attention along the iterations. Based on the class AP result of the privous iteration, we suppose those classes with lower APs should have higher learning complexity and be paid for attention in the subsequent iterations.

## EXPERIMENTAL RESULTS ON IMAGENET

We furthermore carry out experiments on ILSVRC2012 Classification dataset(Russakovsky et al., 2015) which consists of 1.2 million images for training, and 50,000 for validation. There are 1,000 classes in the dataset. For data augmentation and normalization, we adopt scaling, ramdom cropping and horizontal flipping as in (He et al., 2016; Huang et al., 2016). Similar to the experiments on CIFAR-100, the error per class on training datasets approaches zero, we update the category distribution w.r.t the class errors on validation datasets. Since the test dataset of ImageNet are not available, we just report the results on the validation sets, following He et al. (2016); Huang et al. (2016) for ImageNet. When we train ResNet50 networks on ImageNet, the initial learning rates are set to 0.1 and divided by 0.1 at epoch [30, 60]. Similar to (He et al., 2015; Huang et al., 2016) again, we train the network for 90 epoches. The performances of emsembled classifier with different number of base networks are shown in the bottom rows of (table 2). These base ResNet networks with diverse capabilities are combined to generate more discriminative ensemble classifier.

| Datasets | Network | T=1 | T=2 | T=3 | T=4 |
|---|---|---|---|---|---|
| MNIST | MLP | 4.73 | 2.22 | 1.87 | 1.86 |
| CIFAR-100 | ResNet56(He et al., 2016)* | 29.53 | 26.65 | 24.97 | 24.15 |
| | DenseNet-BC(k=12)(Huang et al., 2016)* | 30.78 | 28.95 | 27.60 | 26.64 |
| ImageNet | ResNet50 (He et al., 2015) | 24.18(7.49) | 23.28(6.98) | 22.96(6.81) | 22.96(6.79) |

Table 2: Single crop test error rate(%) along iterations with boosted models. * indicates updating weights with $\lambda = 0.7$, while the others $\lambda = 1$. Blue indicates the use of pre-trained model from TorchVision. () indicates top 5 error rate.

## CONCLUSIONS

In this paper, we develop a deep boosting algorithm is to learn more discriminative ensemble classifier by combining a set of base experts with diverse capabilities. The base experts are from the family of deep CNNs and they are sequentially trained to recognize a set of object classes in an easy-to-hard way according to their learning complexities. As for the future network, we would like to investigate the performance of heterogeneous base deep networks from different families.

## ACKNOWLEDGMENTS

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
