# OpenReview forum: "Deep Boosting of Diverse Experts"
_ICLR.cc/2018/Conference — Reject_

### Official Review · AnonReviewer1 · 2017-11-23
**A boosting with CNNs approach with fixed class weights between iterations - Many fundamental questions and experiments haven't been addressed**

**Rating:** 2
**Confidence:** 5

**Review:**

This paper consider a version of boosting where in each iteration only class weights are updated rather than sample weights and apply that to a series of CNNs for object recognition tasks.

While the paper is comprehensive in their derivations (very similar to original boosting papers and in many cases one to one translation of derivations), it lacks addressing a few fundamental questions:

- AdaBoost optimises exponential loss function via functional gradient descent in the space of weak learners. It's not clear what kind of loss function is really being optimised here. It feels like it should be the same, but the tweaks applied to fix weights across all samples for a class doesn't make it not clear what is that really gets optimised at the end.
- While the motivation is that classes have different complexities to learn and hence you might want each base model to focus on different classes, it is not clear why this methods should be better than normal boosting: if a class is more difficult, it's expected that their samples will have higher weights and hence the next base model will focus more on them. And crudely speaking, you can think of a class weight to be the expectation of its sample weights and you will end up in a similar setup.
- Choice of using large CNNs as base models for boosting isn't appealing in practical terms, such models will give you the ability to have only a few iterations and hence you can't achieve any convergence that often is the target of boosting models with many base learners.
- Experimentally, paper would benefit with better comparisons and studies: 1) state-of-the-art methods haven't been compared against (e.g. ImageNet experiment compares to 2 years old method) 2) comparisons to using normal AdaBoost on more complex methods haven't been studied (other than the MNIST) 3) comparison to simply ensembling with random initialisations.

Other comments:
- Paper would benefit from writing improvements to make it read better.
- "simply use the weighted error function": I don't think this is correct, AdaBoost loss function is an exponential loss. When you train the base learners, their loss functions will become weighted.
-  "to replace the softmax error function (used in deep learning)": I don't think we have softmax error function

---

> ### Public Comment · (anonymous) · 2017-12-22
> **Response**
>
>  We thank the reviewers for their comments. Individual points are addressed below.
>
> -The objective function of our algorithm is the weighted margin between the average correct classification probability and the average incorrect classification probability, as in Eq.(3) and Eq.(4). The first term of Eq.(4) is just the likelihood of positive samples. So, the loss can be seen as the opposite number of objective function Eq.(3). Maximizing the margin in Eq.(3) is equivalent to minimizing the loss. To update the weight distribution for different categories, we employ an exponential updating rule as in Eq.(7), which encourages focusing on the categories that are hard to classify.
>
> -For large-scale visual recognition, it is worth noting that every object class may contain large numbers of hard images due to huge intra-class visual diversity, thus weighting the sample errors may not be able to achieve the same effects as weighting the object classes according to their learning complexities, e.g., weighting the sample errors may not be able to improve the accuracy rates for the hard object classes. Because large numbers of object classes may have different learning complexities, the errors from the hard object classes and the easy ones may have significantly different effects on optimizing their joint objective function. Therefore, it is very attractive to invest new boosting algorithms that can train the deep networks for the hard object classes and the easy ones sequentially in an easy-to-hard way, such that the ensemble network can improve the accuracy rates for the hard object classes at certain degrees while effectively maintaining high accuracy rates for the easy ones.
>
> -Because large numbers of object classes may have different learning complexities, the sample errors from the hard object classes and the easy ones may have significantly different roles in optimizing their joint objective function on learning their joint deep network. Unfortunately, for large-scale visual recognition (i.e., recognizing large numbers of object classes), weighting the sample errors individually (like traditional deep boosting approaches) may not be able to achieve the same effects as weighting the object classes directly according to their learning complexities, e.g., treating the sample errors from the hard object classes and the easy ones to be equally important may easily distract their joint deep network on achieving higher accuracy rates on recognizing the easy object classes but paying less attentions on correcting the sample errors from the hard object classes. Therefore, it is very attractive to invest new boosting algorithms that can train the deep networks for the hard object classes and the easy ones sequentially in an easy-to-hard way, such that the ensemble network can improve the accuracy rates for the hard object classes at certain degrees while effectively maintaining high accuracy rates for the easy ones.

---

### Official Review · AnonReviewer2 · 2017-11-28
**This paper proposed a boosting method for learning an ensemble of neural networks**

**Rating:** 6
**Confidence:** 3

**Review:**

In conventional boosting methods, one puts a weight on each sample. The wrongly classified samples get large weights such that in the next round those samples will be more likely to get right.  Thus the learned weak learner at this round will make different mistakes.
This idea however is difficult to be applied to deep learning with a large amount of data. This paper instead designed a new boosting method which puts large weights on the category with large error in this round.  In other words samples in the same category will have the same weight

Error bound is derived.  Experiments show its usefulness though experiments are limited

---

### Official Review · AnonReviewer4 · 2017-12-06
**deep learning with the boosting trick**

**Rating:** 5
**Confidence:** 4

**Review:**

This paper applies the boosting trick to deep learning. The idea is quite straightforward, and the paper is relatively easy to follow. The proposed algorithm is validated on several image classification datasets.

The paper is its current form has the following issues:
1. There is hardly any baseline compared in the paper. The proposed algorithm is essentially an ensemble algorithm, there exist several works on deep model ensemble (e.g., Boosted convolutional neural networks, and Snapshot Ensemble) should be compared against.
2. I did not carefully check all the proofs, but seems most of the proof can be moved to supplementary to keep the paper more concise.
3. In Eq. (3), \tilde{D} is not defined.
4. Under the assumption $\epsilon_t(l) > \frac{1}{2\lambda}$, the definition of $\beta_t$ in Eq.8 does not satisfy $0 < \beta_t < 1$.
5. How many layers is the DenseNet-BC used in this paper? Why the error rate reported here is higher than that in the original paper?
Typo:
In Session 3 Line 7, there is a missing reference.
In Session 3 Line 10, “1,00 object classes” should be “100 object classes”.
In Line 3 of the paragraph below Equation 5, “classe” should be “class”.

---

> ### Public Comment · (anonymous) · 2017-12-22
> **Response**
>
> We thank the reviewer for the comments. Individual points are addressed below.
> -In Eq. (3), \tilde{D} is not defined.
> A:As described in the paragraph below Eq. (4),  \tilde{D}_t(l) is the normalized importance score for class l in the tth base expert. Its initial value is 1/C. Then it will be updated iteratively as in Eq. (9).
> -Under the assumption $\epsilon_t(l) > \frac{1}{2\lambda}$, the definition of $\beta_t$ in Eq.8 does not satisfy $0 < \beta_t < 1$.
> A:In Section 3.3, we discuss the selection of $\lambda$.   In the case $\epsilon_t(l) > \frac{1}{2\lambda}$，it means that the l-th category is the hard category, and we decrease the hyper-parameter $\lambda$ such that $0 < \beta_t < 1$ holds.
> - How many layers is the DenseNet-BC used in this paper? Why the error rate reported here is higher than that in the original paper?
> A: A 100-layer DenseNet-BC model is used in this paper on CIFAR100 which is the same in the original paper (https://arxiv.org/pdf/1608.06993.pdf). The reason why the error rate here is higher is mainly due to that we do not use all 50,000 samples on training split and validation split at the final run, which is the training trick reported in the original paper. As mentioned in our paper, we only use the validation split for the update of the weights in the following iteration. Another minor factor may be that we only train the model once in the first iteration and do not run many times for selection of the best model since we care more about the effects of our boosting algorithm.

---

### Decision · Program_Chairs · 2018-01-29
**ICLR 2018 Conference Acceptance Decision**

**Decision:**

Reject

**Comment:**

The paper presents a boosting method and uses it to train an ensemble of convnets for image classification. The paper lacks conceptual and empirical comparisons with alternative boosting and ensembling methods. In fact, it is not even clear from the experimental results whether or not the proposed method outperforms a simple baseline model that averages the predictions of T independently trained convolutional networks.